# Phloridzin Reveals New Treatment Strategies for Liver Fibrosis

**DOI:** 10.3390/ph15070896

**Published:** 2022-07-20

**Authors:** Yahong Shi, Tun Yan, Xi Lu, Kai Li, Yifeng Nie, Chuqiao Jiao, Huizhen Sun, Tingting Li, Xiang Li, Dong Han

**Affiliations:** 1School of Life Sciences, Beijing University of Chinese Medicine, Beijing 100029, China; shiyh2019@nanoctr.cn (Y.S.); yantun1985@163.com (T.Y.); lux2020@nanoctr.cn (X.L.); lik2018@nanoctr.cn (K.L.); sunhz2020@nanoctr.cn (H.S.); litt2020@nanoctr.cn (T.L.); 2National Center for Nanoscience and Technology, Beijing 100190, China; nieyf@nanoctr.cn; 3College of Pharmacy, Baotou Medical College, Baotou 014042, China; 4Beijing City International School, Beijing 100022, China; mosheng0624@126.com

**Keywords:** phloridzin, liver fibrosis, mRNA, lncRNA, ferroptosis, energy metabolism, biomechanics

## Abstract

Liver fibrosis is an urgent public health problem which is difficult to resolve. However, various drugs for the treatment of liver fibrosis in clinical practice have their own problems during use. In this study, we used phloridzin to treat hepatic fibrosis in the CCl_4_-induced C57/BL6N mouse model, which was extracted from lychee core, a traditional Chinese medicine. The therapeutic effect was evaluated by biochemical index detections and ultrasound detection. Furthermore, in order to determine the mechanism of phloridzin in the treatment of liver fibrosis, we performed high-throughput sequencing of mRNA and lncRNA in different groups of liver tissues. The results showed that compared with the model group, the phloridzin-treated groups revealed a significant decrease in collagen deposition and decreased levels of serum alanine aminotransferase, aspartate aminotransferase, laminin, and hyaluronic acid. GO and KEGG pathway enrichment analysis of the differential mRNAs was performed and revealed that phloridzin mainly affects cell ferroptosis. Gene co-expression analysis showed that the target genes of lncRNA were obvious in cell components such as focal adhesions, intercellular adhesion, and cell–substrate junctions and in metabolic pathways such as carbon metabolism. These results showed that phloridizin can effectively treat liver fibrosis, and the mechanism may involve ferroptosis, carbon metabolism, and related changes in biomechanics.

## 1. Introduction

With the incidence of liver disease increasing year by year, liver fibrosis and cirrhosis have resulted in serious public health problems [1]. Liver fibrosis is a common outcome of many chronic liver diseases due to the massive accumulation of extracellular matrix (ECM) such as that found in viral hepatitis, alcoholic steatohepatitis, nonalcoholic steatohepatitis, metabolic disorders, and autoimmune and chronic inflammatory conditions [2,3,4,5]. The structural unit of the liver is the hepatic lobule, which is composed of thousands of hepatic sinuses. Sinuses are specific capillaries in the liver which are distributed in a central radial manner within the lobules of the liver, and there is a narrow Disse space between the endothelial cells of hepatic sinuses and liver parenchymal cells which contains hepatic stellate cells and hepatic intrinsic macrophages. There have been many studies on the pathogenesis of hepatic fibrosis, in which hepatic stellate cells are considered to be the central link in the occurrence of hepatic fibrosis. In addition, there are also studies on the causes of liver fibrosis, including damage to hepatocytes [6], fenestrae decrease in liver sinusoidal endothelial cells [7], and activation of liver Kupffer cells [8]. Liver fibrosis is a common pathological stage of virtually all chronic hepatic diseases, and is taken seriously by numerous clinicians [9,10]. Although many treatment strategies and numerous drugs have been used to treat liver fibrosis, few have been shown to have a satisfactory clinical effect. Therefore, the development of effective and specific drugs to improve liver fibrosis is urgently needed.

Plant-derived medicines, from aspirin to artemisinin, play a very important role in the clinic. The introduction of artemisinin has resulted in Chinese medicine receiving significant attention worldwide. From the clinical experience with traditional Chinese medicine (TCM), we can identify specific drugs for the treatment of diseases. Phloridzin is a type of flavonoid which is widely distributed in many plants. Phloridzin was first used in the treatment of diabetes [11]. It is the prodrug of dapagliflozin and canagliflozin. It has powerful but gentle biological activity. In recent years, researchers have found that phloridzin has many other uses, for example, in the treatment of liver cancer [12]. However, the role of phloridzin in treating liver fibrosis has not been well studied. Lychee seed, a TCM rich in phloridzin [13], is widely used in the treatment of hepatic diseases in TCM clinics. Thus, we attempted to determine the potential of phloridzin in the treatment of hepatic fibrosis (Figure 1). Silibinin, a drug commonly used in liver disease, was selected as the positive control. Hematoxylin and eosin (H&E) staining and Sirius red staining were used to observe liver tissue cell morphology and calculate the area of collagen deposition, serum alanine aminotransferase (ALT) and aspartate aminotransferase (AST) content assays were performed to assess liver function, and serum laminin (LN) and hyaluronic acid (HA) content assays were performed to evaluate liver fibrosis ECM deposition. Liver ultrasound was carried out to detect changes in liver density and hardness. Subsequently, mRNA and lncRNA sequencing of liver tissue in different groups was used to investigate the mechanism of phloridzin in the treatment of liver fibrosis.

## 2. Results

### 2.1. Phloridzin Alleviated Liver Fibrosis in the CCl_4_-Induced C57/BL6N Mouse Model

In order to examine the possible effect of phloridzin on liver fibrosis, phloridzin was used to treat the CCl_4_-induced liver fibrosis in a mouse model. The appearance and histological images of normal and fibrotic livers as well as treated livers were observed. The collagen deposition area as well as serum indices and liver density were examined. As revealed by H&E and Sirius red staining, the administration of CCl_4_ led to significant cell damage in the liver as well as an increase in the area of collagen deposition compared to the control group, while phloridzin markedly alleviated liver cell damage and reduced the collagen deposition area to nearly normal levels (Figure 2C,D,G). In addition, serum levels of the liver function markers (ALT and AST) were assessed in treated and untreated animals. The results showed significant differences in ALT and AST levels between treated and untreated animals (Figure 2H,I). Furthermore, serum levels of the liver fibrosis markers (HA and LN) between the different groups were examined using ELISA. The results showed that serum levels of HA and LN decreased significantly after treatment compared with the model group (Figure 2J,K). Taken together, these results indicate that phloridzin is effective in the treatment of CCl_4_-induced liver fibrosis.

### 2.2. Phloridzin Ameliorated Liver Stiffness as Shown by Ultrasound Imaging

Mechanical homeostasis is the ability to generate and repair mechanical forces within tissues and organs to maintain function. When tissue is damaged, mechanical homeostasis changes and a repair procedure is initiated, including the recruitment and activation of intrinsic mesenchymal cells. While tissue fibrosis is the failure of mechanical homeostasis reconstruction during tissue damage repair, biomechanics-related transcriptional and epigenetic mechanisms participate in the default pathway of mechanically sensitive cell activation, resulting in progressive extracellular matrix deposition and tissue destruction. There are also significant mechanical and biomechanical changes in the formation and treatment of liver fibrosis [14]. To examine the changes in biomechanics in vivo, different groups were monitored by hepatic echogenicity, in which the reflection of ultrasound waves represented a change in stiffness which corresponded to the brightness in the ultrasound image as well as the 3D surface plot (Figure 2E). Improvement of fibrosis in the treatment group was seen by decreased intensity of hepatic echogenicity compared with the model group. Statistical analysis of liver brightness in each group showed that compared with the model group, each treatment group had a different degree of reduced intensity (Figure 2F). Among these groups, the results of the silibinin treatment group and the phloridzin high-dose group were significantly different.

### 2.3. Expression Profiles of lncRNAs and mRNAs in Different Groups

The RNA sequencing data (including mRNA and lncRNA) were obtained from four groups (control group, model group, silibinin group, and phloridzin high-dose group) using the Illumina NovaSeq platform. We analyzed differently expressed (DE) mRNAs and lncRNAs with the cutoff LogFC (fold changes) >1 or <−1 along with adjust *p* value < 0.05 and false discovery rate (FDR) < 0.05.

The results showed that there were 4643 DE mRNAs (2758 upregulated and 1885 downregulated) and 2720 DE lncRNAs (1427 upregulated and 1293 downregulated) in the model group compared with the control group, 3272 DE mRNAs (1598 upregulated and 1674 downregulated) and 1930 DE lncRNAs (973 upregulated and 957 downregulated) in the silibinin group compared with the model group, and 3212 DE mRNAs (1420 upregulated and 1792 downregulated) and 1865 DE lncRNAs (799 upregulated and 1066 downregulated) in the phloridzin high-dose group compared with the model group.

### 2.4. GO and KEGG Pathway Analysis Explained the Different Mechanism between Silibinin and Phloridzin

To identify the different mechanisms of different drugs, we analyzed the DE mRNAs between the four groups. DE mRNAs in the samples are shown in a heat map (Figure 3A), bar chart (Figure 3B), and Venn and Venn upset diagrams (Figure 3C,D).

According to the analysis of DE mRNAs previously described, we performed GO and KEGG pathway analysis. The top 10 terms of the GO analysis are shown in Figure 4A–F. Interestingly, GO terms were significantly expressed in biological processes, mainly including “organic acid metabolic process”, “oxoacid metabolic process”, and “carboxylic acid metabolic process”. Differences between the two drugs were seen in the KEGG pathway enrichment analysis, which are shown in Figure 4G–J. Differences between silibinin and the model group were mainly in the signaling pathways related to the TNF signaling pathway, ErbB signaling pathway, and cell cycle. However, the DEG mRNAs in the phloridzin-treated group mainly focused on amino sugar and amino acid metabolism, ferroptosis, PPAR signaling pathway, and carbohydrate metabolism, suggesting that phloridzin can treat liver fibrosis by affecting energy metabolism or protein synthesis.

### 2.5. lncRNA–mRNA Interaction and lncRNA Target Gene Prediction

To determine the mechanism of phloridzin in the treatment of liver fibrosis in detail, we mined the sequencing data and analyzed the co-expression of DE lncRNAs and mRNAs. On the basis of the lncRNA–mRNA relationship pair obtained from the co-expression analysis, the presence of targeted regulation was predicted by lncTar software [15]. A visual regulatory network for the lncRNA–mRNA relationship was drawn in target gene prediction results by Cytoscape 3.7.2 and is shown in Figure 5A. We then performed GO and KEGG pathway analysis to assess the biological significance of these target prediction genes (Figure 5B,C). The GO analysis indicated enrichment in some cellular components related to the ECM and biomechanics, such as cell–substrate adherens junction, focal adhesion, adherens junction, and anchoring junction. Additionally, the KEGG pathway enrichment analysis showed that the target mRNAs of DEG lncRNAs were mostly concentrated in pathways related to amino acid metabolism and carbon metabolism, which are associated with energy metabolism.

## 3. Discussion

The increasing incidence of liver disease has led to an increase in mortality year by year, and has gradually become an important cause of human death worldwide. Liver fibrosis is a common process in various liver diseases, such as non-alcoholic fatty liver disease, alcoholic fatty liver disease, viral hepatitis and so on. The study of treatment at the liver fibrosis stage is important and urgent. Researchers continue to explore therapeutic measures for liver fibrosis, including extracting compounds from natural plants to develop new drugs [16], and identifying the causes and therapeutic targets of liver fibrosis [8,17,18,19,20]. In this study, a single compound, phloridzin, was extracted from lychee seed to determine its mechanism of action in the treatment of liver fibrosis. By analyzing the experimental results, we found that phloridzin significantly improved the degree of collagen deposition in liver fibrosis, reduced tissue stiffness, reduced the levels of ALT and AST in serum, and reduced the levels of HA and LN in serum. Some of these indexes have the same tendency as in the literature [21]. In addition, in order to reflect the change contributed to phloridzin to liver hardness and density of liver fibrosis, we also carried out liver ultrasound imaging experiments. This indicator is crucial for the detection of liver fibrosis in vivo. In order to more deeply explore the mechanism of phloridzin in the treatment of liver fibrosis, as well as explore more ideas for the treatment of hepatic fibrosis in the future, we carried out high-throughput sequencing of livers in different groups of livers, and explored the mechanisms of phloridzin and silibinin in the treatment of liver fibrosis from the perspective of epigenetics. Analysis of the mRNA sequencing results of livers in the different groups revealed that there were significant differences in the pharmacological mechanisms of phloridzin and silibinin, a commonly used hepatoprotective drug in the clinic. The mechanism of silibinin in liver fibrosis is mostly concentrated on the TNF signaling pathway and the ErB signaling pathway. It is speculated that it may have an estrogen-like effect, which is consistent with literature reports [22,23]. The mechanisms of phloridzin in liver fibrosis are mostly concentrated on ferroptosis, the PPAR signaling pathway, and carbohydrate metabolism. The results reveal some novel therapeutic strategies and research ideas.

“Ferroptosis” was coined in 2012 to describe an iron-dependent regulated form of cell death caused by the accumulation of lipid-based reactive oxygen species, which was different to apoptosis, necrocytosis, and autophagy [24]. Ferroptosis is caused by the accumulation of iron ions, energy metabolism, and lipid peroxidation [25]. Therefore, regulation of ferroptosis is a new method for the treatment of cell death or excessive cell proliferation-related diseases. The role of ferroptosis in liver fibrosis has received considerable attention [19]. Studies have shown that hepatic fibrosis can be treated by inducing the death of activated hepatic stellate cells by regulating ferroptosis [26,27]. Ferroptosis is closely related to energy metabolism. Energy metabolism, as a regulator of ferroptosis [28], has received more and more attention, such as in tumors [29]. The relationship between ferroptosis and liver fibrosis caused by energy metabolism is unclear. The results of this study may provide patients with a new treatment for liver fibrosis, by regulating energy metabolism and inducing ferroptosis.

Changes in cell viability inevitably lead to changes in cell secretory function. Through target gene prediction of differential lncRNAs, we also found that among the genes closely related to lncRNA regulation, a number of genes related to biomechanics such as ECM and cell adhesion are involved. Liver fibrosis is a pathological process closely related to biomechanics [30,31]. More and more studies have been conducted on the mechanics of fibrotic liver, and treatment strategies for the liver’s mechanical environment are gradually being developed [17,32]. LncRNAs are longer than 200 nucleotides and do not appear to code for proteins [33]. Studies have shown that lncRNAs play an important role in many life activities, such as the dose compensation effect [34], epigenetic regulation [35], cell cycle regulation [36], and cell differentiation regulation [37], and have become a hotspot in genetics research. Interestingly, when we looked for target genes in differential lncRNAs and in differential mRNAs, ECM-related genes were highlighted. The ECM component is a non-cellular component secreted by cells. It belongs to the interstitial tissue and has no secretion and expression function. It supports and connects cells and even tissues, and transports some substances. LncRNAs are also regulatory RNAs that do not function as expressed proteins. This led us to consider whether lncRNAs also belong to the interstitial tissue components in genetic material, and whether non-coding RNAs are closely related to interstitial components such as the ECM as well as a non-secretory function such as biomechanics. Some scholars have also noted a connection between the two. The correlation between lncRNA and the ECM has been examined in osteoarthritis [38,39]. There are also reports showing that lncRNAs are closely related to the secretion of ECM proteins [40]. However, there are few studies on the correlation between lncRNA and biomechanics. Our findings provide data on this correlation. However, the exact close correlation requires further study. These are also innovative points in our work: using high-throughput sequencing technology for the first time to reveal the complex mechanisms of phloridzin in the treatment of liver fibrosis, and combining energy metabolism, ferroptosis, lncrna, and biomechanics to explore new ways to treat liver fibrosis.

## 4. Materials and Methods

### 4.1. Animals

Forty-eight healthy male C57/BL6N mice (weighing 20 ± 2 g) were obtained from Beijing Vitong Lihua Co., Ltd. (Beijing, China). The mice were divided into 12 cages with a regulated environment (12-h dark/light cycle, 22 ± 2 °C temperature and 50 ± 5% humidity). Experimental procedures were in accordance with the International Ethical Guidelines for Animal Care and were approved by the Ethical Committee, National Center for Nanoscience and Technology, Beijing, China.

### 4.2. Drugs, Chemicals and Reagent Kits

Phloridzin and silibinin were purchased from Meryer Chemical Technology Co., Ltd. (Shanghai, China); mouse LN, HA, and ELISA kits were purchased from CLOUD-CLONE CORP. (CCC, Houston, TX, USA).

### 4.3. Methods

#### 4.3.1. Experimental Design

The mice were randomly divided into 6 groups: the control group, model group, silibinin group (27.3 mg/kg), phloridzin low-dose group (10 mg/kg), phloridzin middle-dose group (20 mg/kg), and phloridzin high-dose group (40 mg/kg). Each group contained 8 mice.

Liver fibrosis in the mouse model was induced by intraperitoneal injection of carbon tetrachloride (CCl_4_) dissolved in olive oil (1:9, *v*/*v*) at a dosage was 5 mL/kg body weight twice a week for 8 weeks. Drug therapy was started on the fifth week. After 8 weeks, the eyeballs of the mice were removed for blood collection after anesthesia with isoflurane. After that, the mice were sacrificed with isoflurane and the livers excised.

#### 4.3.2. Histological Staining

Liver tissues were fixed with 4% paraformaldehyde and processed for H&E staining and Sirius red staining to evaluate the degree of fibrosis.

#### 4.3.3. Analysis of Serum Biochemical Parameters

Serum was collected following centrifugation of whole blood cell samples at 3000 rpm at 4 ℃ for 15 min. Serum ALT and AST levels were analyzed by fully automatic clinical analyzers (Hitachi-7100, Tokyo, Japan). Serum LN and HA levels were determined using ELISA kits according to the manufacturer’s instructions based on the principle of the competitive inhibition enzyme immunoassay technique.

#### 4.3.4. Ultrasound Imaging

The stiffness and density of the liver were monitored by high-frequency ultrasound imaging using a Visual Sonic Vevo 2100 system as previously described [41]. Briefly, the mice were anesthetized with 2.5% isoflurane and the abdominal fur was removed. Ultrasonic coupling gel was applied to the mouse skin after fixing the mice on the Vevo mouse-handling table in the supine position. Mice were scanned from the ventral body wall using the MS55D real-time MicroScan transducer and the Vevo 2100 imaging system. The livers were imaged in the parasternal long axis view and three measurements of the portal vein area were obtained. The brightness of each echohepatograph was quantified by integrated density using Image J 1.53e.

#### 4.3.5. RNA-Seq and Computational Analysis

RNA-Seq analysis was performed using the Illumina NovaSeq platform. The process included five steps: (i) extraction of total RNA from mouse liver tissue; (ii) construction of a cDNA library; (iii) RNA sequencing and screening; (iv) differentially expressed mRNA and lncRNA analysis; (v) different expression gene (DEG) mRNA and DEG lncRNA correlation analysis.

The R package DEseq was used to evaluate DEG mRNA and DEG lncRNA. GO analysis and KEGG pathway analysis were performed using the OmicShare tools, a free online platform for data analysis [42]. The lncRNAs-targeted mRNAs and mRNAs were combined and the lncRNA-mRNA regulatory network was visualized using Cytoscape 3.7.2.

### 4.4. Statistical Analysis

All data are presented as means ± standard deviation. The statistical significance of differences among the groups was evaluated by one-way analysis of variance (ANOVA), followed by multiple comparisons between individual groups using the LSD correction. Differences were considered significant for *p* < 0.05.

## 5. Conclusions

In this study, we used phloridzin to treat hepatic fibrosis and revealed the pharmacological mechanisms of phloridzin. The results showed that phloridzin can regulate liver function indexes; extracellular matrix components; and the stiffness of fibrotic liver through energy metabolism, ferroptosis, biomechanics, and other pathways. This work represents the first time these results and mechanisms have been mentioned in phloridzin medicinal study. Furthermore, new therapeutic ideas based on the regulation of energy metabolism and ferroptosis and lncRNA regulation of biomechanics have been proposed herein. These results provide a reference for future research on the treatment of liver fibrosis.

## Figures and Tables

**Figure 1 pharmaceuticals-15-00896-f001:**
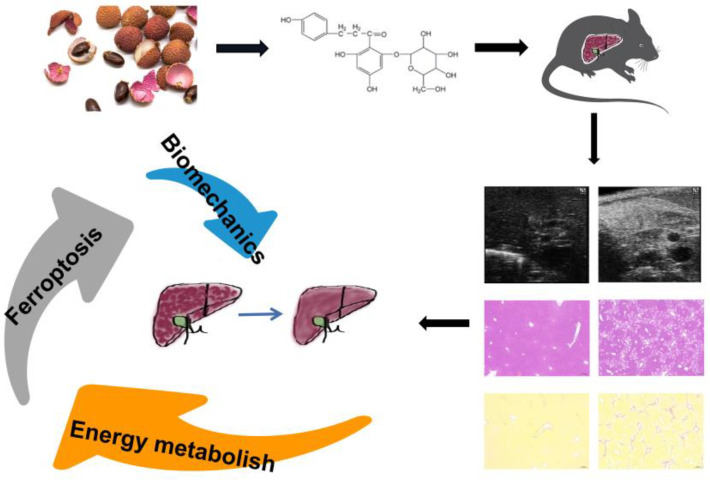
The compound phloridzin derived from lychee seed in the treatment of liver fibrosis.

**Figure 2 pharmaceuticals-15-00896-f002:**
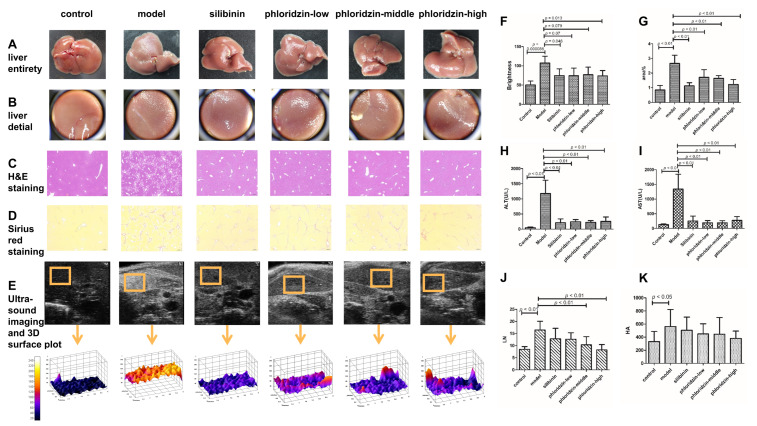
Pharmacodynamic results of phloridzin in the treatment of liver fibrosis. (**A**) Representative photos of liver appearance in each group. (**B**) Detailed view of liver appearance in each group. (**C**) Histological images of liver in each group after H&E stain. (**D**) Histological images of liver in each group after Sirius red stain. (**E**) Representative in vivo ultrasound imaging of livers from different groups. Fibrosis improvement can be seen by a decreased intensity in hepatic echogenicity. The 3D surface plots of the ultrasound images within the orange-lined squares correspond to the echogenic uniformity in the liver. (**F**) Statistics of the liver brightness of each group shows that compared with the model group, each treatment group has different degrees of reduction. Among them, the results of the silibinin treatment group and the high-dose phloridzin group are significantly different. (**G**) Percentage of area occupied by fibers in Sirius red-stained sections of liver in each group. (**H**,**I**) AST and ALT levels of each group were detected. (**J**,**K**) LN and HA levels of each group were detected by ELISA.

**Figure 3 pharmaceuticals-15-00896-f003:**
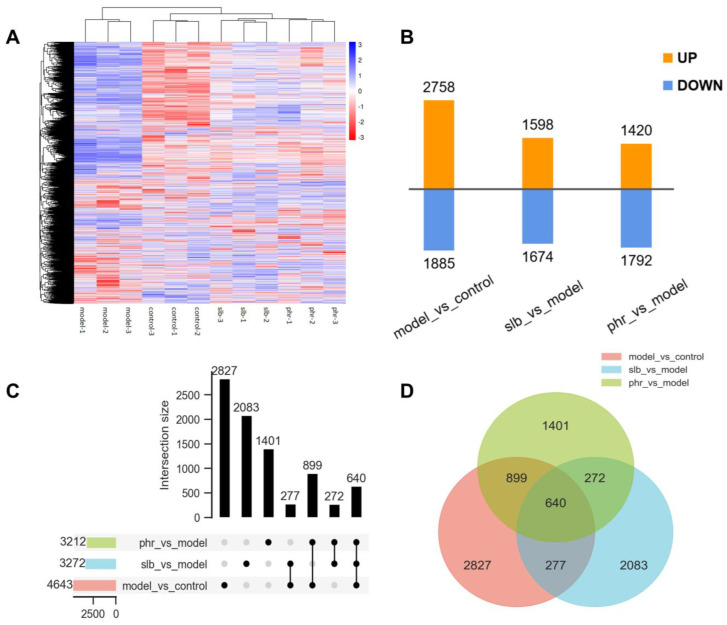
The statistics of DE mRNAs between control group vs model group, silibinin group vs. model group, and phloridzin vs. model group. DE mRNAs in the samples are shown using a heat map (**A**), bar chart (**B**), Venn upset diagram (**C**), and Venn diagram (**D**).

**Figure 4 pharmaceuticals-15-00896-f004:**
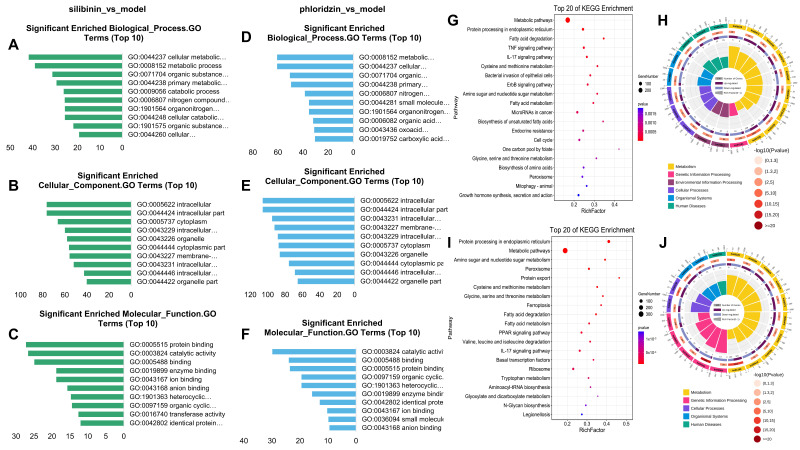
GO and KEGG pathway analysis explain the different mechanism between silibinin and phloridzin. (**A**–**C**) Top 10 terms of GO enrichment analysis of DEG mRNAs between silibinin group and model group. (**D**–**F**) Top 10 terms of GO enrichment analysis of DEG mRNAs between phloridzin group and model group. (**G**,**H**) Top 20 terms of KEGG pathway enrichment analysis of DEG mRNAs between silibinin group and model group. (**I**,**J**) Top 20 terms of KEGG pathway enrichment analysis of DEG mRNAs between phloridzin group and model group.

**Figure 5 pharmaceuticals-15-00896-f005:**
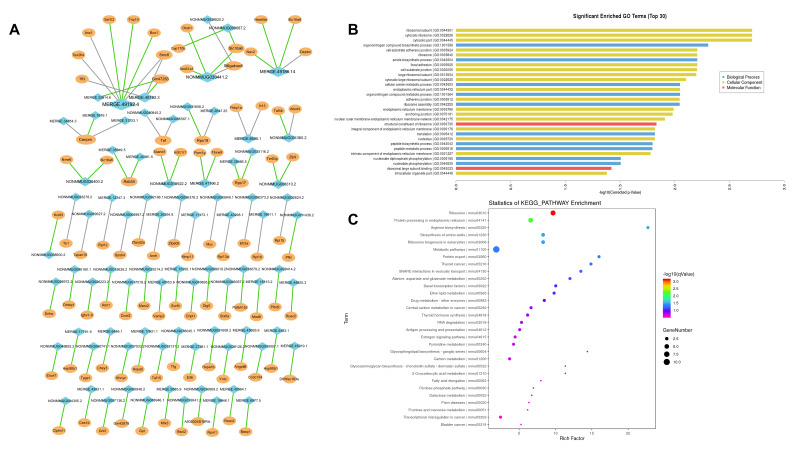
lncRNA-mRNA interaction and lncRNA target gene prediction. A visual regulatory nework for the lncRNA-mRNA relationship was drawn from target gene prediction results by Ctoscape 3.7.2, shown in (**A**). Then, we preformed the GO and KEGG pathway analysis based on the predicted mRNAs. (**B**) shows the top 30 terms of GO analysis and (**C**) shows the top 30 terms of KEGG pathway analysis.

## Data Availability

The data are contained within the article. New high-throughput sequencing (HTS) datasets (RNA-seq) have been deposited into the GEO database (GSE205540).

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
