# Peer review of "Phloridzin Reveals New Treatment Strategies for Liver Fibrosis"

_pharmaceuticals, 2022, doi:10.3390/ph15070896_

Round 1
Reviewer 1 Report
Authors have investigated the potential of Phloridzin isolated from Lychee seed as a therapeutic modality to treat fibrosis. The results have been compared with silibinin flavolignan. The experimental protocol and interpretation of data is well presented, however, there is room for improvement. Following suggestions must be addressed/incorporated in the revised version.
1. What dose of silbinin was used in this study? It will be easier for readers to compare the efficacy of phloridzin versus silibinin.
2. What was the dose selected for low, middle and high phloridzin treated mice groups?
3. How many mice were there in each group. Was it 8/per gp? Mention in the experimental part.
4. Which statistical test was used for multiple comparisons?
5. Conclusion should be elaborated.
6. Fig 2: Correct the figure legend of fig 2B; Liver detial to Liver detail
7. Languague needs thorough revision. Some typos/syntax errors/grammatical errors are given below;
a. Line 10: "Liver fibrosis....... Resolve". This sentence in the abstract is ambiguous. Rephrase it to convey the right meaning.
b. Line 12: CCl4; 4 should be subscript.
c. line 16-17: Showed has been used twice in one sentence, please rephrase it.
d. Line 44: It is better to replace doctors with clinicians.
e. "liver fibrosis mouse model", add ïn"fibrosis.
Reviewer 2 Report
Despite the interesting topic and the large amount of analyzes performed, the work must be subject of many corrections. In many cases, the documentation is unreadable.
Figure 2C - unreadable image
Figure 2D - unreadable image
Figure 4 - unreadable image
Figure 5 - unreadable image
line 287 - please describe the method of animal euthanasia
line 292 - how and where blood was taken for analysis
All references should include DOI in the description
